# Thermodynamic Multi-Field Coupling Optimization of Microsystem Based on Artificial Intelligence

**DOI:** 10.3390/mi14020411

**Published:** 2023-02-09

**Authors:** Guangbao Shan, Xudong Wu, Guoliang Li, Chaoyang Xing, Shengchang Zhang, Yu Fu

**Affiliations:** 1School of Microelectronics, Xidian University, Xi’an 710071, China; 2Beijing Institute of Aerospace Control Devices, Beijing 100039, China; 3China Academy of Aerospace Standardization and Product Assurance, Beijing 100071, China

**Keywords:** TSV, optimization, particle swarm optimization algorithm, microsystem

## Abstract

An efficient multi-objective optimization method of temperature and stress for a microsystem based on particle swarm optimization (PSO) was established, which is used to map the relationship between through-silicon via (TSV) structural design parameters and performance objectives in the microsystem, and complete optimization temperature, stress and thermal expansion deformation efficiently. The relationship between the design and performance parameters is obtained by a finite element method (FEM) simulation model. The neural network is built and trained in order to understand the mapping relationship. Then, the design parameters are iteratively optimized using the PSO algorithm, and the FEM results are used to verify the efficiency and reliability of the optimization methods. When the optimization target of peak temperature, bump temperature, TSV temperature, maximum stress and maximum thermal deformation are set as 100 °C, 55 °C, 35 °C, 180 Mpa and 12 μm, the optimization results are as follows: the peak temperature is 97.90 °C, the bump temperature is 56.01 °C, the TSV temperature is 31.52 °C, the maximum stress is 247.4 Mpa and the maximum expansion deformation is 11.14 μm. The corresponding TSV structure design parameters are as follows: the radius of TSV is 10.28 μm, the pitch is 65 μm and the thickness of SiO_2_ is 0.83 μm. The error between the optimization result and the target temperature is 2.1%, 1.8%, 9.9%, 37.4% and 7.2% respectively. The PSO method has been verified by regression analysis, and the difference between the temperature and deformation optimization results of the FEM method is not more than 3%. The stress error has been analyzed, and the reliability of the developed method has been verified. While ensuring the accuracy of the results, the proposed optimization method reduces the time consumption of a single simulation from 2 h to 70 s, saves a lot of time and human resources, greatly improves the efficiency of the optimization design of microsystems, and has great significance for the development of microsystems.

## 1. Introduction

With the continuous shrinking of CMOS process nodes, the continuation of Moore’s Law is restricted by physical limits. As an extension of Moore’s Law, three-dimensional integrated circuits have received more and more attention [1]. Three-dimensional microsystem (3D microsystem) technology has the advantages of miniaturization, integration, intelligence, low cost, high performance, mass production and so on, and is widely used in various fields [2,3,4]. Because of the increasing power density in a smaller area and the higher thermal resistance, the temperature distribution on the chip increases significantly in the stacking of a 3D microsystem, and the thermal problem becomes particularly serious. On the one hand, high temperature will affect the performance of the devices in the circuit. On the other hand, the thermal stress caused by the mismatch of thermal expansion coefficients between different materials will also have a negative influence on the circuit [5,6,7]. Excessive thermal stress will bring serious thermal stress problems to the complex three-dimensional integrated structure, causing TSV copper pillars to peel off the contact surface of the substrate, or cracks in the bonding, micro bumps, and sink bottom [8]. Even if there is no destruction to device structure, the reduction of device carrier mobility caused by thermal stress will also lead to the deterioration of circuit performance [9,10]. Therefore, reducing the thermal stress in the silicon substrate is particularly important to ensure the reliability of the device. Thermal and stress coupling problems caused by heat accumulation in microsystems is attracting more and more attention. Therefore, it is important to investigate the thermal effect on microsystems and develop a fast and efficient microsystem design method.

In the last decade, the thermal models of TSV and TSV arrays have been systematically investigated and established. Meanwhile, the thermal models of 3D microsystems have also been studied. The thermal models in 3D microsystems mainly include 3D networks, equivalent thermal conductivity (ETC) models, FEM, and so on [11]. Lua J H [12] deduced an empirical formula for thermal conductivity under different TSV structural parameters based on modeling simulation. Chen et al. [13] established and verified the equivalent thermal resistance model of TSV, and rapidly predicted the thermal performance of a 3D stacked die package based on TSV through the proposed model. Xiao et al. [14] established a fast and accurate equivalent thermal model for TSV; several parameters, including pitch, the thickness of SiO_2_, and the radius of TSV, have been considered, and the accuracy of the proposed model has been verified by FEM. Chen Z [15] simulated the thermo-mechanical reliability of TSV-based stacked packaging by means of FEM analysis, and analyze the data to optimize structural parameters by design of experiment (DOE) method.

The heat generation and heat dissipation performance of each module in the micro system are the two major factors that determine the temperature characteristics of the 3D integrated system. Therefore, most researches consider to alleviate the power problem caused by the heat accumulation of the micro system from the aspects of enhancing the heat dissipation performance and reducing the heat generation. Kuan H. Lu et al. [16] studied the thermomechanical reliability of 3D interconnects and found that the thermal stress in silicon decreases with the increase of the distance from the isolated TSV, and increases with the increase of the TSV diameter. Hanjie Yang et al. [17] established a multi -field coupling analysis model to study the influence of the SiO_2_ insulation layer, structural parameters and insulating layer materials on the thermal stress of TSV. They proposed that the SiO_2_ insulation layer has an important influence on the thermal stress of TSV. The greater the thickness of the SiO_2_ insulation layer with low thermal conductivity, the smaller the maximum equivalent stress value of the SiO_2_/Cu interface and the equivalent stress value of Si area, and the smaller the structural parameter q (q = D/P), the smaller the maximum equivalent stress and maximum deformation are, and BCB and polyethylene terephthalate can effectively reduce the thermal stress. Yen Yi Germany Hoe [18] analyzed the influence of TSV spacing, TSV width-to-height ratio, oxide layer thickness, and TSV material changes on the thermal distribution and maximum value of TSV, and analyzed the combination of different combination factors on the thermal effect of TSV in integrated circuits, but there was a certain deviation because no coupling analysis was used. Previous research shows that the most effective methods proposed at present to optimize thermal stress include increasing heat sinks to improve heat dissipation, changing TSV size filling materials, and adopting new structures to reduce thermal stress.

In the past, people mostly used some traditional methods to change TSV size structure for optimization such as FEM or ETC models. These traditional methods were too complicated and depended on expert experience [19,20]. The thermal stress coupling optimization problem of 3D microsystems is extremely complex, involving multiple physical fields such as electricity, heat, force, etc. The coupled multiple physical fields greatly increase the calculation amount of the FEM model. For example, when FEM simulation software is used for simulation design, a pair of simple TSV electric thermal multi-field coupling simulations takes about 20 min. When considering the accuracy, it will take several hours to select a more refined mesh generation finish simulation. The structural parameters of the microsystem are modified repeatedly for iterative optimization, which wastes a lot of time and greatly reduces the design efficiency of the microsystem. In another example, human experience judgment, analysis and calculation are inevitably wrong, especially when the amount of calculation becomes cumbersome and complex, which puts forward more stringent requirements on the ability of designers. In terms of time resources and labor costs, it is more and more difficult to design complex microsystems with multi-field coupling and multi-objective cooperative optimization, so it is necessary to find a fast and efficient method to improve the efficiency of temperature optimization in microsystems. In recent years, artificial intelligence (AI) algorithms have gradually emerged. These algorithms learn from human experience include genetic algorithms, ant colony optimization algorithms, and PSO algorithms, etc., which are widely used in electronics, automation, management and other industries to improve work efficiency and effect [21,22]. In 2011, Pervaiz, et al. successfully combined artificial intelligence to apply a particle swarm optimization algorithm to medical disease detection [23]. In 2013, Chen, et al. applied particle swarm optimization to the field of chemistry, proposed a new hybrid gradient particle swarm optimization (HGPSO) algorithm, and completed the challenging dynamic optimization problem in chemistry [24]. In 2022, Saini, et al. used an AI algorithm to study and optimize the availability of biological and chemical units in sewage treatment plants to achieve the required level of reliability and maintainability [25]. In 2021, Lim, et al. took an experimental study on the vertical squeezing route Taylor flow for θ = 20, 45, 90, 135, and 160° at different flow rates of helium (He) and ethanol. By employing an extreme θ of 20 and 160°, an in-depth knowledge of the associated mechanics is gained [26]. Using machine learning instead of expert experience, artificial intelligence methods have unique advantages in solving problems with complex computation and huge time consumption. The neural network model is trained by a large amount of data to establish the mapping relationship, which is faster and less prone to error than manual judgment. AI methods have great potential in realizing multi-field coupling and multi-objective co-operative optimization of complex microsystems.

In this research, a fast and efficient design method based on artificial intelligence is proposed for the complex thermal problems of microsystems. The optimization process of thermal problems in microsystems based on the AI method is introduced in Section 2. The thermal optimization results of the microsystem are obtained in Section 3. In Section 4, the speediness and coincidence of the optimization results are verified by comparing with the FEM simulation. Section 5 concludes this paper.

## 2. Materials and Methods

In order to optimize the structure of TSV to solve the thermal and stress problems problem, the intelligent optimization method for its structure parameters is developed based on COMSOL software and AI methods, as shown in Figure 1. Based on the obtained data, the neural network models are trained to describe the relationship between the TSV structure parameters and performance parameters of microsystems. According to the established optimization criteria, the PSO algorithm is used to optimize the design parameters of TSV.

The details of the developed method are as follows:

Step 1 Obtaining data by FEM. FEM is a numerical technique for solving approximate solutions of boundary value problems of partial differential equations. It uses the variation method to minimize the error function and produce a stable solution. Compared with traditional methods such as thermal resistance network and equivalent model, the finite element method not only has a higher calculation accuracy, but also can adapt to various complex shapes. Therefore, it has become an effective engineering analysis method and has been widely used in the research and design of microsystems. In order to obtain the database required for neural network training, the TSV array model based on FEM needs to be accurately described. First of all, a 4 × 4 TSV array is simulated and discussed. The establishment and simulation steps of the finite element model are shown in Figure 2 [3,27]. Figure 3 shows the structural model between the two TSVs. TSV parameters are R (TSV radius), P (TSV pitch) and t_ox_ (oxide liner thickness). Since this is a simulation, the dimensions here are taken as representative data. The thermal conductive layer is 20 μm, the BCB layer is 10 μm, and the silicon substrate is 50 μm. After setting up the above parts, the next part is the core TSV part. The TSV composed of copper and bump provide vertical interconnection through silicon substrate. Due to the need for subsequent experiments, multiple sets of values are taken for the dimensional parameters of the TSV. The TSV diameters are taken as 3–11 μm, pitches as 25–65 μm, and the thickness of SiO_2_ as 0.1–0.9 μm. The Cu_3_Sn micro bumps are simplified as cylinders with a radius of 6 μm and height of 10 μm. The thermal multi-field coupling of solid heat transfer and thermal expansion is selected as the physical field. In order to make the heat flow into the model uniformly, a 50 μm × 50 μm heating chip with a thickness of 10 μm and a heat dissipation rate of 1 W is placed in the center of the model [28,29], and the thermal conductivity of the thermal conductivity layer between the chip and the PCB is set to 400 W/mk. The bottom surface of the model is set as a fixed constraint, the temperature is 293.15 K at room temperature, and the remaining surfaces except for the above-mentioned top surface and bottom surface is set as thermal insulation.

After the material of the geometry is defined, the pad and TSV filling material are set to copper, the substrate material to silicon, the micro bump material to Cu_3_Sn, and the insulating layer material of the TSV to silicon dioxide, as shown in the simulation [30,31]. The materials used and their related physical parameters are shown in Table 1 [32]. The calculation speed and accuracy are both taken into account in the mesh division. The ultra-fine mesh division is selected at the interface of different materials, and the fine mesh division is considered at the non-critical parts.

The purpose of this experiment is to reduce the temperature and stress by optimizing the structural parameters of the TSV to ensure the normal operation of the system. In the microsystem, the peak temperature caused by thermal problems cannot be higher than the limit temperature. The bonding layer between different layers of the system may break and peel under high temperature; the key connections between different chip layers, such as bump and TSV, are also significantly affected by temperature. Therefore, in this study, while the peak temperature $T_{p}$, maximum stress $S_{t}$, and thermal expansion $D_{s}$ are optimized, the bump temperature $\;\;T_{c}$ and TSV temperature $\;T_{t}$ are also selected as the optimization goal. The three factors selected in this section that affect the temperature of the TSV array are: TSV radius R, TSV spacing P and insulating layer thickness $t_{ox}$. Refer to the TSV structure manufacturing process to select three different values for the above three factors as their factor levels, as shown in Table 2. According to the determined three test factors and their nine levels, the orthogonal table L81(9^3^) is used to arrange the test, and 81 different TSV array structure parameter combinations are obtained.

There are many factors affecting deformation in finite element modeling and simulation. Temperature and strain rate are two important factors that affect the stress–strain curve. For most materials, the higher the temperature, the softer the material, that is, the smoother the stress–strain curve. In this paper, in order to build a neural network model database, a finite element model for thermal stress analysis is established, and uses steady-state simulation under ideal conditions.

TSV array model and temperature simulation result are shown in the Figure 4.

As shown in Figure 5, the change of temperature and stress is obviously affected by those parameters. With the changes of various parameters, the changes of dependent variables are irregular, it is difficult to summarize their laws through simple mathematical models. Therefore, the optimization method combined with AI proposed in this paper has a certain prospect.

Step 2 Establishing Neural Network Models for TSV optimal design BP network is the core part of the feedforward neural network, but there are some defects, such as slow learning convergence, no guarantee of convergence to the global minimum point, and uncertain network structure. In this paper, a genetic algorithm is used to optimize the BP neural network to find the most appropriate initialization weight and deviation value, so that it can easily converge to the global optimal solution. The main steps include coding, fitness calculation, selection, genetic operation, etc. The database is divided based on orthogonal experiment, and 70% of data in the database are taken as training data, 15% as verification data, and the remaining 15% as test data. Means square error (MSE) is used to represent the performance. The GA genetic algorithm is used to optimize the BP neural network. The main ideas to realize the BP-GA neural network are: BP neural network determination, GA optimization of weight threshold and BP training prediction.

The neural networks are trained by the obtained data. The inputs of neural network are R, P and $t_{ox}$, while the outputs are $T_{p}$, $T_{c\;}$, $T_{t}$,$\;D_{s}$ and$\;S_{t}.\;$Based on the database, the models about TSV design parameters and optimization target-mapping relationship are established.

Taking the peak temperature as an example, the neural network optimization process is shown in Figure 6. In neural network training, the change of fitness with genetic algebra and the distribution of predicted output and expected output in test samples are shown in Figure 7. With the increase of genetic algebra, the optimal value of fitness gradually approaches the ideal value. Among the 61 groups of results obtained, the predicted range of peak temperature is 92.10–180.20 °C, and the actual range obtained is 92.57–180.18 °C. The predicted output in the figure is highly consistent with the expected output, and the error between each group of predicted and actual output is not more than 0.5%, reflecting the good computational performance of the neural network model.

Step 3 Establishing Optimization Criteria. The optimization criteria are established based on the performance parameters, including peak temperature$\;T_{p}$, bump temperature $T_{c}$, $\mathrm{TSV}\;\mathrm{temperature}\;T_{t}$, $\mathrm{thermal}\;\mathrm{stress}\;\;S_{t}$ and$\;\mathrm{expansion}\;\mathrm{deformation}\;D_{S}$. The optimization criteria J can be mathematically expressed as
(1)$$J=\alpha (T_{p}-T_{p}{}_{des}{)}^{2}+\beta {\left(T_{c}-T_{c}{}_{des}\right)}^{2}+\gamma {\left(T_{t}-T_{tdes}\right)}^{2}+\lambda {\left(S_{t}-S_{tdes}\right)}^{2}+\eta {\left(D_{s}-D_{sdes}\right)}^{2}$$
where des is the abbreviation of designed, different optimization target values are selected based on the actual process and working conditions of the microsystem, and the values of the optimization goals are as follows, the values of $T_{pdes},\;T_{cdes}$, $\;T_{tdes},\;S_{tdes}\;$and $D_{sdes\;}$are 95, 55, 35,180 and 12. The size of these values can also be modified according to the actual situation. α, β, γ, λ and η are weight coefficients of$\;T_{p}$, $\;T_{c}$, $\;T_{t}$, $\;S_{t}$ and $D_{s}$. The size of these weight coefficients is determined manually based on the priority of each optimization objective. In microsystems, in order to solve the thermal mismatch problem caused by heat accumulation due to high power density, the distribution of peak temperature, TSV temperature and bump temperature in the system, as well as the deformation and thermal stress problems caused by thermal expansion and thermal mismatch, are focused in optimization. In the compromise consideration of collaborative optimization of multi-objective problems, the priority of peak temperature is the highest, and its weight coefficient is set to 0.35. The bump temperature, TSV temperature and thermal expansion deformation weight coefficient are set to 0.2, and the thermal stress weight coefficient is set to 0.05.

Step 4 Optimizing the structure parameters by PSO Algorithm Because the PSO algorithm with linear decreasing inertial weight has excellent global and local searching ability, it is adopted in the developed method, and it can be expressed as
(2)$$v_{i}(t+1)=w(iter)v_{i}(t)+c_{1}r_{1}(p_{i}-x_{i}(t))+c_{2}r_{2}(p_{g}-x_{i}(t))$$
(3)$$x_{i}(t+1)=x_{i}(t)+v_{i}(t+1)$$
(4)$$w(iter)=(\frac{iter_{\max}-iter}{iter_{\max}}w_{\max}-w_{\min})+w_{\min}$$
where w is the criteria weight; $p_{i}\;$and$\;p_{g}\;$are best previous positions of it particles and global particles.$\;r_{1}$ and$\;r_{2\;}$are random numbers between [0, 1];${\;c}_{1\;}$and$\;c_{2\;}$are weights of$\;p_{i}$ and$\;p_{g}$; $iter_{\;}$and$\;iter_{\max\;}$are the number and the maximum number of iterations;$\;w_{\min}$ and $w_{\max}\;$are the minimum and maximum of the inertia weight. The process of using the PSO algorithm with linearly decreasing inertia weight to optimize the design parameters of the TSV array in the three-dimensional microsystem is shown in Table 3.

## 3. Results

This paper implements PSO based on the Matlab software platform. First, the structural design parameters are initialized, then the trained neural network and the calculation results of the neural network model are loaded. The next step is to initialize the PSO algorithm parameters and realize the algorithm optimization. Finally, the optimal solution after completing the cycle iteration is outputted. According to the TSV array design parameters, the constructed neural network model is used to predict the optimization target. According to the constructed TSV array peak temperature multi-objective optimization function, the PSO algorithm is used to optimize the TSV array design parameters. Determine whether to obtain the optimal TSV array design parameters: if yes, the intelligent optimization of the peak temperature is completed; otherwise, go back and repeat the above steps. In order to reduce the random error of the PSO algorithm, the optimized design strategy developed has been run independently, 30 times. It takes 70 s to complete the intelligent iteration using the PSO algorithm, while it takes at least 2 h to calculate the target performance corresponding to a group of structural parameters using traditional FEM simulation. When the TSV array model becomes more complex, the consumption of time resources will further increase. As shown in Figure 8, in the final optimization design, the radius of TSV is 10.28 μm, the pitch of TSV is 65.00 μm, the thickness of SiO_2_ is 0.83 μm. The final optimization results are as follows: the global maximum temperature is 97.90 °C, and the error is 2.1% compared with the target 100 °C. The temperature of bump is 56.01 °C and the error is 1.8% compared with the target 55 °C. The temperature of TSV is 31.52 °C, and the error is 9.9% compared with the target 35 °C. The maximum thermal stress is 247.4 Mpa, and the error is 37.4% compared with the target 180 Mpa. The maximum thermal deformation is 11.14 μm, and the error is 7.2% compared with the target 12 μm. It can be seen from the optimization results that there is a large error between the optimization results of thermal stress and the preset target, and the other optimization results are good. The error analysis will be discussed below.

## 4. Discussion

As a mature commercial finite element simulation software, COMSOL’s reliability has been recognized by the industry and widely used in various fields. As an example, in 2021, Lim, et al. studied the hysteresis effect of solutions containing different ionic species and significant pH differences on the system through experiments and finite element simulation [33]. The effectiveness of the developed method is verified by COMSOL software. The TSV array model is built by COMSOL software, and the optimization results are shown in the Table 4. Experiment is an important method to check whether the conclusion of simulation is accurate. However, due to some conditions constraints, multiple regression is used instead of experiment to test the optimization results in this paper. We imported 81 sets of orthogonal data into MATLAB software for regression analysis, and the parameter model obtained is as follows:(5)$$T_{p}=247.611-28.431R-0.280P-1.119t_{ox}+1.394R^{2}-0.011RP+0.005P^{2}$$
(6)$$T_{c}=204.286-24.675R-0.492P-1.246t_{ox}+1.212R^{2}-0.009RP+0.004P^{2}$$
(7)$$T_{t}=55.034-0.063R-0.597P-0.200t_{ox}-0.013R^{2}+0.004P^{2}$$
(8)$$D_{s}=39.755-3.503R-0.295P-0.152t_{ox}+0.167R^{2}+0.001RP+0.002P^{2}$$
(9)$$\begin{array}{l} S_{t}=792.619-5.200R-3.380P-1053.5t_{ox}+1103.9t_{ox}{}^{2}-29.534Rt_{ox}\\ +2.296Pt_{ox} \end{array}$$

Regression analysis results are also shown in Table 4. The optimization results based on the AI method are compared with the results of finite element simulation and multiple regression analysis, and their error conditions are analyzed and compared, and the significance of the proposed AI method is discussed. The structural parameters in the optimization results based on the AI method, namely the TSV radius R is 10.28 μm, the TSV spacing P is 65 μm, and the oxide layer SiO_2_ thickness t_ox_ is 0.83 μm, are respectively substituted into the established TSV array finite element model and multiple regression analysis equation based on COMSOL software, and the performance target results are shown in Table 4. If the result of finite element simulation is taken as the reference value, the peak temperature is 97.98 °C, the peak temperature obtained by the AI method is 97.90 °C, the error is about 0.08%, and the peak temperature obtained by the multiple regression method is 97.30 °C, the error is about 0.69%. The bulge temperature of the finite element simulation results is 57.22 °C, and the bulge temperature obtained based on the AI method is 56.01 °C, with an error of about 2.11%. The bulge temperature obtained by multiple regression is 57.22 °C, with an error of about 1.08%. The temperature of the copper column in the finite element simulation results is 31.49 °C, the temperature of the copper column based on the AI method is 31.52 °C, the error is about 0.10%, and the temperature of the copper column obtained by multiple regression is 30.94 °C, and the error is about 1.75%. When taking the maximum thermal expansion deformation as the optimization target, the maximum thermal expansion deformation obtained by finite element simulation is 11.10 nm, the maximum thermal expansion deformation obtained by the AI method is 11.14 nm, with an error of about 0.36%, and the maximum thermal expansion deformation obtained by multiple regression is 11.21 nm, with an error of about 1.00%. The maximum thermal stress obtained by the AI method is 247.4 Mpa, with an error of about 10.08%. The maximum thermal stress obtained by multiple regression is 256.07 Mpa, with an error of about 6.93%.

From the above comparison, it can be seen that the optimization of the power problem of the microsystem is completed based on the artificial intelligence method proposed in this paper. The overall results obtained by the optimization method are a good fit with the finite element simulation results. Compared with the data prediction obtained by the regression analysis mathematical method, it is closer to the finite element simulation results in terms of peak temperature, copper column temperature and maximum thermal expansion deformation.

In this research, the developed optimization method is based on AI methods. The neural network models play the key roles in the developed method, which is based on the data obtained by COMSOL software. So, the accuracy of the simulation data is very important for the optimization of parameters.

The main sources of errors are model errors and data measurement errors. FEM regards the solution domain as composed of many small interconnected subdomains called finite elements, and assumes an appropriate approximate solution for each element, and then deduces the total satisfaction conditions for solving this domain, so as to obtain the solution of the problem. This solution is not an exact solution, but an approximate solution When using the finite element simulation, it is often necessary to compromise the accuracy and speed. In order to improve the calculation speed, reduce the mesh refinement to a certain extent, which will affect the accuracy of the simulation results. In contrast, the AI method has both efficiency and accuracy, which is its main advantage. On the one hand, in FEM simulation, lots of time and memory will be wasted because of too-fine mesh division in finite element simulation. In order to improve the calculation speed, part of the mesh is ultra-refined, which reduces the accuracy of the results. In order to reduce its error, the accuracy of the data should be taken as the primary condition, and a more precise mesh division should be considered while reasonably improving the FEM model. On the other hand, the data obtained by simulation calculation usually retain two decimal places, and the resulting error is inevitable. By the way, the optimal design of the TSV array is a complex multi-objective collaborative optimization, and it needs to consider various performance parameters. The weight coefficients of different goals are set according to the priority of different optimization goals; it will also affect the optimization results. In this research, the weight coefficient of thermal stress is low, resulting in a large error of thermal stress. To improve the accuracy of stress optimization results, its weight coefficient should be improved.

In this study, the AI method is used to optimize the microsystem, realize the multi-objective collaborative optimization under the multi-field coupling, and find the optimal structural design parameters to optimize the heat and stress, which is much faster than the traditional FEM method. In actual microsystems, more and more complex performance parameter tradeoffs often need to be considered. Compared with the set goal and FEM simulation results, the optimal value of thermal stress in the optimization results of this study still has some errors, which may require other algorithms to improve the stress optimization.

## 5. Conclusions

An intelligent optimization method based on AI is proposed to quickly adjust the structural design parameters of TSV in microsystems and optimize the thermal mechanical coupling problem The main conclusions can be summarized as follows:

(1)Based on the data simulated by COMSOL software, the neural network models are established to characterize the relationship between the design parameters and performance parameters.(2)The PSO algorithm is used to optimize the radius of TSV, the pitch of TSV, and the thickness of the insulating layer, and the optimized parameters are outputted. Calculated by neural network models, the optimized structure parameters and the calculated performance parameters are outputted as the optimization results. The optimization results obtained are as follows: the radius of TSV is 10.28 μm, the pitch of TSV is 65.00 μm, the thickness of SiO_2_ is 0.83 μm, the peak temperature is 97.90 °C, the temperatures of bump and TSV are 56.01 °C, and 31.52 °C. The maximum thermal stress is 247.40 Mpa, the maximum thermal expansion deformation is 11.14 μm. The difference between them and the preset goal are 2.1%,1.8%,9.9%,37.4% and 7.2%.(3)The PSO method is verified by regression analysis, and the difference between the optimization results and FEM method is about 3%. The maximum error of stress shall not exceed 10.07%. The time required for a single simulation has changed from 2 h to 70 s, and the overall efficiency has increased thousands of times. Therefore, the method proposed in this paper is an efficient and accurate TSV array optimization method.

Nowadays, with the development of multi-functional microsystems, the structure is increasingly complex, and the problem of heat accumulation is increasingly prominent. At the same time, the multi-field coupling calculation involved has become huge. Compared with the traditional manual method, the intelligent optimization method proposed in this paper has greatly saved manpower and time, and optimized the heat and stress problems. At the same time, this method is applied to the multi-field coupling and multi-objective collaborative optimization of microsystems. The establishment of appropriate neural network model and the setting of optimization criteria can further solve the collaborative optimization problem between thermal and electrical signals, which is of great significance to the research and development of microsystems.

## Figures and Tables

**Figure 1 micromachines-14-00411-f001:**
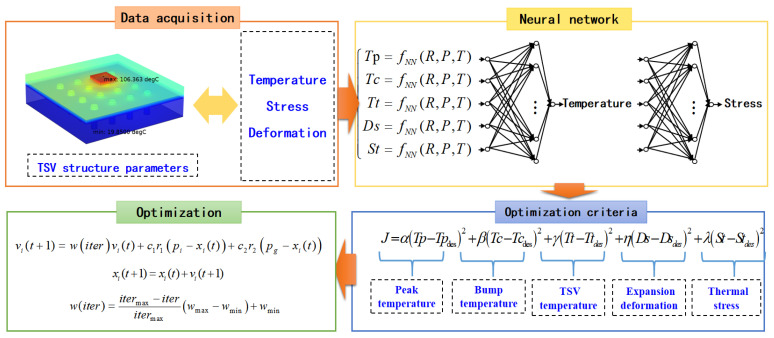
Flowchart of the developed intelligent optimization method for the parameters of TSV.

**Figure 2 micromachines-14-00411-f002:**
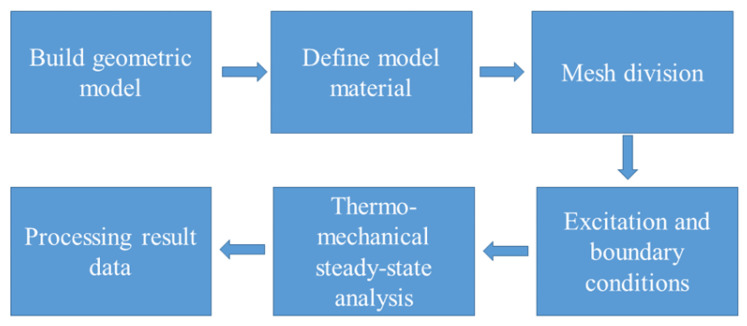
FEM modeling flow chart.

**Figure 3 micromachines-14-00411-f003:**
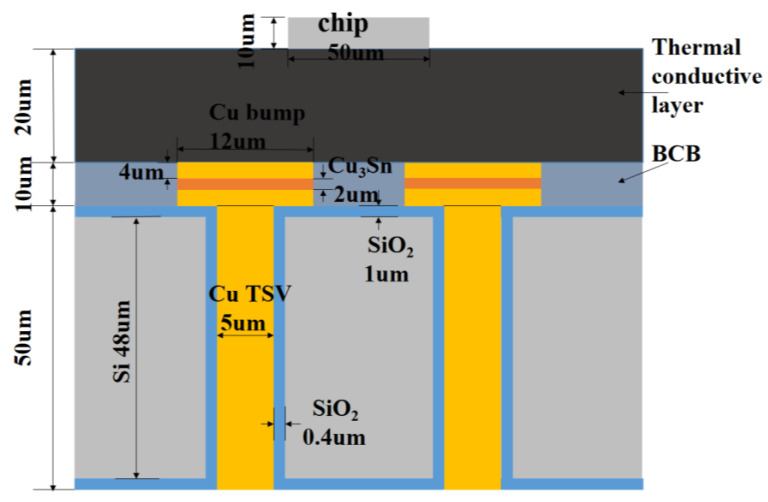
Schematic diagram of TSV structure parameters.

**Figure 4 micromachines-14-00411-f004:**
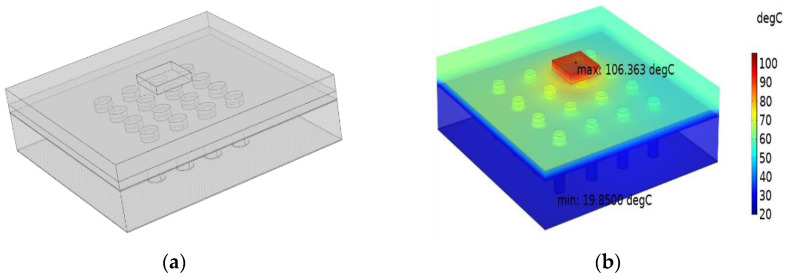
TSV array model and simulation result diagram: (**a**) TSV array model diagram; (**b**) TSV Array Peak Temperature Simulation Results.

**Figure 5 micromachines-14-00411-f005:**
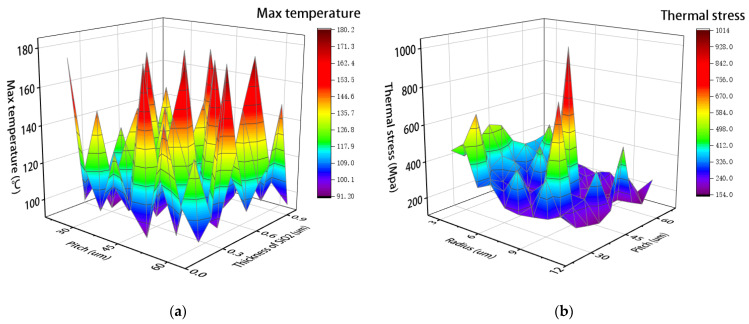
Diagram of performance variation with structural parameters: (**a**) Effect of parameters on peak temperature of TSV; (**b**) Effect of parameters on stress of TSV.

**Figure 6 micromachines-14-00411-f006:**
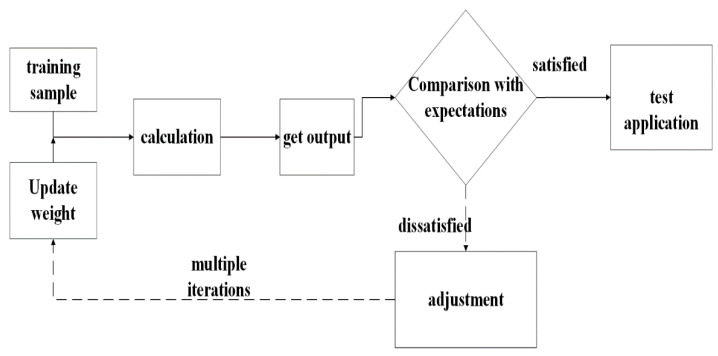
Neural network optimization process.

**Figure 7 micromachines-14-00411-f007:**
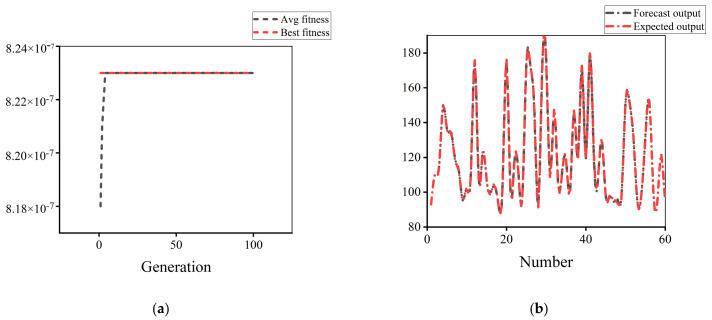
Neural network training graph: (**a**) Fitness variation diagram; (**b**) Test output error diagram.

**Figure 8 micromachines-14-00411-f008:**
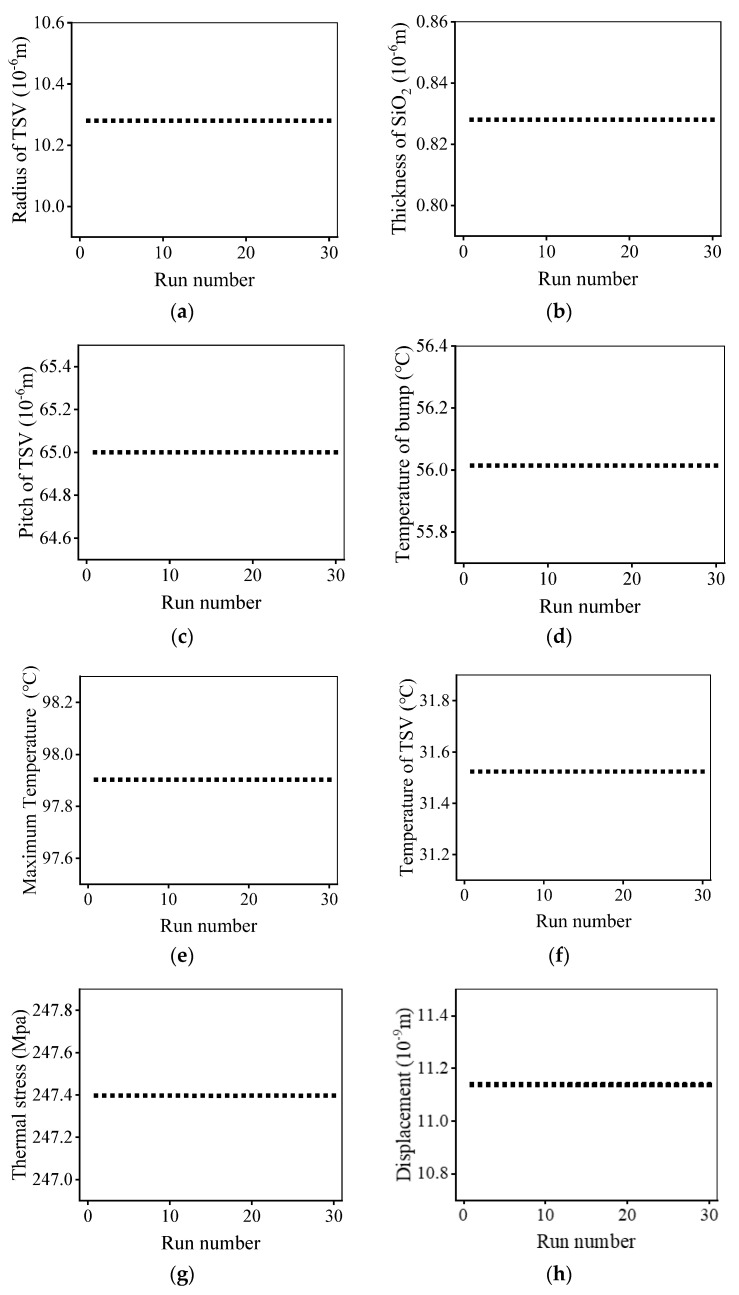
Optimization results: (**a**–**c**) design parameters; (**d**–**h**) optimized goal.

**Table 1 micromachines-14-00411-t001:** Materials and related physical parameters.

Material	SiO_2_	BCB	Cu	Si
Thermal Conductivity (W/m·k)	1.4	0.3	401	130
Density (Kg/m^3^)	2200	1050	8960	2329
Heat capacity (J/(Kg·K))	730	2128	384	700
Dielectric constant (F/m)	4.2	2.65	1.0	11.7
Young’s modulus (Gpa)	72	3	220	130
Poisson’s ratio	0.16	0.34	0.35	0.28
Thermal expansion coefficient (ppm/°C)	0.6	40	18	2.3

**Table 2 micromachines-14-00411-t002:** The parameter combination factor level.

Level	1	2	3	4	5	6	7	8	9
Radius (μm)	3	4	5	6	7	8	9	10	11
Pitch (μm)	25	30	35	40	45	50	55	60	65
Thickness of SiO_2_	0.1	0.2	0.3	0.4	0.5	0.6	0.7	0.8	0.9

**Table 3 micromachines-14-00411-t003:** Intelligent optimization and COMSOL optimization results.

Constant Parameters	Inertia Weight Range	Maximum Iterations	Population Size	Position Range	Velocity Range
*c*_1_ = 2, *c*_2_ = 2	*W* ∈ [0.4, 0.9]	*iter*_max_ = 50	N = 30	*x*_1_ ∈ [3, 11] *x*_2_ ∈ [25, 65] *x*_3_ ∈ [0.1, 0.9]	*v*_1_ ∈ [−1, 1] *v*_2_ ∈ [−5, 5] *v*_3_ ∈ [−0.1, 0.1]

**Table 4 micromachines-14-00411-t004:** Comparison of results of various methods.

Result	Peak Temperature (°C)	Bump Temperature (°C)	TSV Temperature (°C)	Stress (Mpa)	Deformation (mm)
Optimization	97.90	56.01	31.52	247.40	11.14
Comsol	97.98	57.22	31.49	275.13	11.10
Regression	97.30	56.60	30.94	256.07	11.21

## Data Availability

Not applicable.

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
