# Peer review of "Thermodynamic Multi-Field Coupling Optimization of Microsystem Based on Artificial Intelligence"

_micromachines, 2023, doi:10.3390/mi14020411_

Round 1

Reviewer 1 Report

Please see attached PDF comments and suggestions for authors.

Reviewer 2 Report

1. Have you consider thermal conductivity layer in your modeling? Also, in general, it should have two thermal interface materials between chip and heat sink tank. These factors should be added in your FEM model.

2. Please describe more details related your FEM modeling process.

3. The deformation will be influence by many factors such as moisture adsorption, you need consider these in your FEM modeling.

4. The temperature in your modeling should be higher than 100 to consistent with the realistic chip manipulation temperature.

5. When you get your conclusion, you may can do at least one group of experiments to support your modeling data.

Round 2

Reviewer 1 Report

The authors have well-addressed the comments and suggestions. The paper is now ready for publication. Good jobs authors; congrats!